# Formation Mechanism of Aluminide Diffusion Coatings on Ti and Ti-6Al-4V Alloy at the Early Stages of Deposition by Pack Cementation

**DOI:** 10.3390/ma12193097

**Published:** 2019-09-23

**Authors:** Hailiang Du, Ning Tan, Li Fan, Jiajie Zhuang, Zhichao Qiu, Yanhua Lei

**Affiliations:** 1College of Mechanical and Electronic Engineering, Shanghai Jian Qiao University, Shanghai 201306, China; 18126@gench.edu.cn (L.F.); nhezzjj17@163.com (J.Z.); 2Institute of Marine Materials Science and Engineering, College of Ocean Science and Engineering, Shanghai Maritime University, Shanghai 201306, China; tanning986@163.com (N.T.); qzc4321@163.com (Z.Q.)

**Keywords:** diffusion coating, aluminide, pack cementation, Ti, Ti-6Al-4V, mechanism

## Abstract

The diffusion coatings were deposited on commercially pure Ti and Ti-6Al-4V alloy at up to 1000 °C for up to 10 h using the pack cementation method. The pack powders consisted of 4 wt% Al (Al reservoir) and 4 wt% NH_4_Cl (activator) which were balanced with Al_2_O_3_ (inert filler). The growth kinetics of coatings were gravimetrically measured by a high precision balance. The aluminised specimens were characterised by means of scanning electron microscopy (SEM), energy dispersive spectrometer (EDS) and X-ray diffraction (XRD). At the early stages of deposition, a TiO_2_ (rutile) scale, other than aluminide coating, was developed on both materials at <900 °C. As the experimental temperature arose above 900 °C, the rutile layer became unstable and reduced to the low oxidation state of Ti oxides. When the temperature increased to 1000 °C, the TiO_2_ scale dissociated almost completely and the aluminide coating began to develop. After a triple-layered coating was generated, the coating growth was governed by the outward migration of Ti species from the substrates and obeyed the parabolic law. The coating formed consisted of an outer layer of Al_3_Ti, a mid-layer of Al_2_Ti and an inner layer of AlTi. The outer layer of Al_3_Ti dominated the thickness of the aluminide coating.

## 1. Introduction

Pack cementation is regarded as a versatile and economical process frequently employed in energy, aerospace and other industries to make aluminide diffusion coatings on nickel-base superalloys or iron-based alloys [1,2,3,4]. However, there has been an increasing industrial demand to develop this generic technology to allow depositions of diffusion coatings on titanium-based alloys, which would provide long-term stability for the alloys at high temperatures in oxidative and corrosive environments [5,6,7,8,9,10,11]. The formation of a continuous α-Al_2_O_3_ scale on the high Al-containing coating (i.e., Al_3_Ti) deposited by pack cementation and its barrier function to slow down the inward diffusion of oxygen into the substrates attribute to the improved oxidation behaviours of the Ti-based alloys [12,13,14,15,16,17]. Achieving these objectives would bring substantial technological and economic advantages to the aerospace and power generation industries, enabling the full exploitation of low weight but high-specific strength properties of Ti-based alloys at high temperatures of operation.

In general, the pack cementation falls into the category of chemical vapour deposition process [18,19] which is activated by halide salts, for instance, AlCl_3_, NH_4_Cl, NH_4_F, etc. The substrates to be deposited are embedded into a well-mixed powder pack composed of pure or alloyed depositing elements (e.g., Al, Si, Cr, etc.), a halide salt acting as an activator and an inert filler (normally Al_2_O_3_). The whole pack is then heated in a range of high temperatures (600–1200 °C) in an inert gas atmosphere, vacuum or under reducing conditions. The activator reacts, at elevated temperature, with the depositing elements to develop a series of gaseous halides containing the depositing elements. These halide vapours transport through the porous powder mixture to the substrate surface. The coating is generated via dissociation of these halide vapours on the substrate surface and subsequent solid-state interdiffusion between the coated elements and the substrate. 

The additions of alloying elements to the substrate materials not only influence their oxidation behaviours, but also affect the microstructure and oxidation resistance of the aluminide coatings deposited on the Ti-based alloys [6,14,17,20,21]. It is reported [20,21] that the addition of Nb or Cr may facilitate the partial conversion of Al_3_Ti from a tetragonal structure to high symmetric cubic L12 structure, thereby leading to better toughness than the original Al_3_Ti due to the dissolution of Nb or Cr into Al_3_Ti. The change of the brittle tetragonal DO22 structure to the ductile cubic L12 also enhances the oxidation resistance of Al_3_Ti coating. Basuki et al. [17] reported the positive effect of Zr and Y on the oxidation behaviour of aluminised TiAl alloys by pack cementation.

The coating processes play an important role in generating the coating microstructures [12,22,23]. The high Al activity of the powder pack at low temperatures usually results in a single layer of Al_3_Ti and no other intermetallic compounds showed up and no solid solution of A1 in the substrate alloy formed [22]. This type of coating is usually required to be annealed at higher temperature. The in-pack process produced a single-layer of Al_3_Ti whilst the out-pack method gave rise to a double layered coating of an outer Al_3_Ti layer and an inner Al_2_Ti layer on-TiAl alloy [23]. Due to lack of contact between the substrate and powder in the out-pack process, the surface finish and purity of the coatings were improved so that the oxidation resistance increased [12]. Tsipas and Gordo [24] successfully co-deposited Mo and Al on Ti and Ti-6Al-4 V alloy using the one-step pack cementation process and found the development of a thin layer of TiN, AlN and (Ti,Al) N. However, it appears that the information with regard to the early stages of aluminisation process on the Ti-based alloys are missing.

In this paper, the aluminisation kinetics and the influence of temperatures on the coating deposition process at the early stages are investigated in order to establish the formation mechanism of aluminide diffusion coatings on Ti and Ti-6Al-4V alloy by the pack cementation process.

## 2. Materials and Methods

### 2.1. Materials and Specimens

The bar materials with 20 mm diameter of commercially pure Ti and Ti-6Al-4V alloy were supplied by Jiangsu Wanqian Gongpin Network Technology Co. Ltd. Jiangyin, Jiangsu, China. The specimens had a thickness of 1.5 mm and were cut by an electrical discharge machine (EDM). The specimens were ground on SiC emery papers up to 1000 grit finish. The ground samples were washed in tap water and then degreased in acetone in an ultrasonic cleaner for 20 m. The dimension and weight of the cleaned specimens were measured and recorded and then put into the sample bags individually. The aluminium powder (99.9 wt% purity, 38–75 μm) as the Al reservoir in the pack cementation process was purchased from Aladdin Chemical Industries (Shanghai, China). NH_4_Cl (AR > 99.5%) was the activator and Al_2_O_3_ (150–250 μm) powder was the inert filler; both were supplied by Shanghai Titan Scientific Co., Ltd. (Shanghai, China).

### 2.2. Aluminising

Both Ti and Ti-6Al-4V alloy were aluminised by means of the pack cementation process. The pack powders consisted of 4 wt% Al, 4 wt% NH_4_Cl and 92 wt% Al_2_O_3_; they were thoroughly mixed and then loaded in an alumina crucible. The specimens to be coated were embedded into the powder mixture and the crucible was screwed up using an alumina lid. The sealed crucible containing the pack powders and specimens were then uploaded into a horizontal silica tube which was contained in an electrical furnace. The silica tube was fitted with gas circulation fittings so that the inert gas, argon, could be introduced into the tube. After the silica tube was sealed, Ar began to flush the tube and the furnace was heated up at a rate of 10 °C/min and kept at 150 °C for 4 h to push air out of the silica tube. At the same time, any moisture remaining in the pack powder mixture evaporated and exited out of the crucible and silica tube with the flowing Ar. The furnace was further heated up at the same heating rate up to aluminising temperatures, remained for certain periods of time and cooled down at a rate of 2 °C/min. During the whole period of the aluminising process, Ar continued to flush through the silica tube. When the furnace cooled down to room temperature, the silica tube was opened, and the crucible was unloaded from the furnace. The aluminised specimens were brushed using a soft brush and gently ground on the SiC emery paper of 1500 grit to remove the stuck residual powders. The samples were then washed in water and ultrasonically cleaned in acetone and the weights were measured again. The selected samples for the cross-section examination were mounted in Bakelite, ground on emery papers down to 1500 grit finish and then polished using diamond paste of 6 and 1 μm. 

In order to investigate the early stages of aluminisation, the specimens were heated-up to certain temperatures and then cooled down immediately. Table 1 illustrates the programme for the aluminising experiments.

### 2.3. Characterisation

The cleaned aluminised Ti and Ti-6Al-4V alloy were initially observed by use of optical microscopy (OP, Leica DMI 3000M, Wetzlar, Germany) and the detailed surface and cross-sectioned morphologies were investigated using scanning electron microscopy (SEM, EM30PLUS, COXEM, Daejeon, Korea). The chemical compositions and elemental distribution in the cross-sectioned specimens were characterised by energy-disperse spectrometry (EDS, Oxford Instrument, Abingdon, England), that was equipped to the SEM instrument. The phase and crystalline features of the aluminised samples were analysed by X-ray diffraction (XRD) (X’Pert Pro, Netherlands Analytical Company, Almelo, Netherlands) with monochromatic Cu-Kα radiation (λ = 1.54178 A). XRD diffraction patterns were acquired with 2θ in the range of 20–90° at a scan rate of 1°/min, using a voltage of 40 kV and a current of 100 mA. 

## 3. Results

### 3.1. Kinetics

In order to investigate the initial stages of the aluminising process, the furnace was heated up to 800, 850, 900, 950 and 1000 °C and then cooled down at a cooling rate of 2 °C/min respectively. In addition, the furnace was kept at these temperatures for the duration of 0 h. It was found that a scale formed at 800 °C was very fragile and partially spalled off when the samples were unloaded from the crucible. As a result, it was difficult to measure the weight of samples. From 850 °C, the adhesion between the scale and substrate was improved and therefore, it became possible to measure the weight of the samples. The weight gains for pure Ti at 850, 900, 950 and 1000 °C for 0 h heating are depicted in Figure 1. On one hand, the samples were stuck with residual powders from the crucible which needed to be brushed off hard. On the other hand, heavy cleaning may cause spallation of the scale. Therefore, both actions must be carefully balanced. It was noticed that the weight changes were slightly fluctuated, which might be attributed to the cleaning process. However, it appears that the weight gains did not increase during the period of heating up from 850 to 1000 °C.

Figure 2 reveals the weight gains as a function of aluminising time at 1000 °C. During the first hour of the aluminising process, the weights of both pure Ti and Ti-6Al-4V alloy increased moderately. Afterwards, the weight gains followed the parabolic pattern, indicating that the aluminising process was controlled by the elemental diffusion. The weight gains for the pure Ti appears larger than for the Ti-6Al-4V alloy. The influence of alloying elements became more pronounced with prolonging the exposure time.

### 3.2. Compositions and Microstructures

The aluminised Ti and Ti-6Al-4V alloy were analysed by scanning electron microscopy (SEM), energy dispersive spectrometer (EDS) and X-ray diffraction (XRD). Figure 3 shows SEM morphology and EDS mapping for the cross-sectioned Ti aluminised at 800 °C for 0 h. It should be pointed out that severe spallation of the film formed on the sample which probably occurred within the cooling period. The cross-sectioned sample was prepared with great care. It can be seen that a large crack developed between the film and the Ti substrate and titanium and oxygen were enriched in the scale. The XRD profile, as illustrated in Figure 4, surprisingly demonstrates that the scale formed on the surface of aluminised Ti was dominated by TiO_2_ (rutile), rather than Al-Ti intermetallic compounds. The XRD results also show the presence of Al_2_O_3_ which is surely the residual powder of Al_2_O_3_ from the aluminising pack. Due to the scale spallation, the Ti substrate was exposed and the Ti peaks were observed on the XRD profile. With increasing aluminising temperature to 900 °C, only TiO_2_ (rutile) was chiefly observed. When the aluminising temperature further arose to 950 °C, new forms of titanium oxides, predominately Ti_1.86_O_3_, developed, as revealed in Figure 5, indicating that the rutile which formed at lower temperatures was gradually reduced with temperature. Also, other low-oxidation state Ti oxides (e.g., TiO_1.04_, TiO_0.48_) were identified. This is further confirmed by the SEM and EDS results, as depicted in Figure 6. In fact, the rutile phase was reduced at the scale/substrate interface. At top of the scale, the domination of TiO_2_ still prevails. These results demonstrate that the reduction of the rutile phase began at the scale/substrate interface. In addition, the adherence of scale to the substrate appears much improved and only a few voids are observed. When the aluminising temperature reached 1000 °C, a further reduced form of Ti_2_O was identified, as illustrated by XRD analysis (Figure 7). The TiO_2_ was still traceable whilst the titanium mono-aluminide (TiAl) emerged on the specimen surface, as indicated in the Figure 7. 

The final aluminising temperature was kept at 1000 °C for various durations of 1, 3, 6 and 10 h. Figure 8 shows the SEM and EDS results for the aluminised Ti-6Al-4V at 1000 °C for 1 h. A coating of triple-layers was developed. The outer layer consisted of the aluminide phase of high Al content and the aluminides of relatively low Al concentration composed the mid-layer and inner layer. The XRD results, as given in Figure 9, confirmed that Al_19_Ti_6_, Al_3_Ti and Al_2_Ti were unambiguously identified in the coating. The presence of Al_19_Ti_6_ indicated the high Al concentration built up on the sample surface. The AlTi phase was not found in the XRD pattern, probably because the X-ray was unable to the reach the phase. The thicknesses of the outer layer, mid-layer and inner layer were around 7 (±0.5), 3 (±0.3) and 5 (±0.5) μm, respectively. By prolonging the aluminising time, the thickness of the outer layer was increased significantly whilst the growth of the mid-layer and inner layer were suppressed. Figure 10 depicts the SEM morphology and EDS mapping after the Ti-6Al-4V alloy was aluminised at 1000 °C for 6 h. The coating appears relatively even, and the thickness became approximately 40 (±1) μm.

Figure 11 gives the SEM image and point measurement of elements of the aluminised Ti at 1000 °C for 10 h. It is apparent that the coating became thicker and three layers developed. The quantitative analyses have demonstrated that the outer layer, mid-layer and inner layer were composed of Al_3_Ti, Al_2_Ti and AlTi, respectively. It should also be pointed that a significant amount of oxygen (14.3 wt%) was dissolved in the Ti substrate. It is believed that the dissolved oxygen comes from the dissociation of titanium of oxides. However, the coating was uneven and a wave-like interface between the coating and substrate is observed. The surface morphologies of the deposited coatings for 6 h of aluminisation carried out by SEM for the Ti and Ti-6Al-4V alloy, are shown in Figure 12. A rough surface is observed on the aluminised Ti whilst the relatively smooth morphology was formed on the aluminised Ti-6Al-4V alloy. The residual Al_2_O_3_ powders remained on the surface, particularly for the aluminised Ti.

## 4. Discussion

In the classical aluminisation process by pack cementation, particularly on the Ni- and Fe-based alloys, the metallic powder of Al reacts with an activator (e.g., NH_4_Cl) and forms gaseous aluminium chlorides which transport to the surface of the component to be coated, where the Al chlorides interact with the sample surface and dissociate into the Al atoms and the chlorine anions. The freed Al atoms diffuse into the substrate surface zone to generate a Ni aluminide coating on the Ni-based alloy or a Fe aluminide coating on the Fe-based alloy via interdiffusion. The driving force for the interdiffusion is the activity or concentration gradient between the environment which has the diffusing elements such as Al and the surface of the components (i.e., Ni or Fe). However, the initial stages of the deposition process, particularly during the heating up period, has been overlooked and the formation mechanisms of the aluminide coating have never been studied. 

The experimental results reported in this paper appear to be never recorded in the literature, particularly, at the early stage of the aluminisation process. In this study, it was found that when Ti and Ti-6Al-4V alloy were aluminised in a powder pack containing 4 wt% Al and 4 wt% NH_4_Cl balanced with Al_2_O_3_, a titanium oxide scale of about 20 μm thickness, rather than a titanium aluminide coating, was developed during the heating up period. The titanium oxide scale predominately consisted of rutile TiO_2_ at lower temperatures (e.g., at <900 °C). When the aluminising temperature rose above a certain temperature (i.e., >900 °C), the high oxidation state of TiO_2_ began to be reduced to lower oxidation states. Even at 1000 °C, a traceable amount of TiO_2_ was still identified. It was found that, only at this temperature, the titanium mono-aluminide (AlTi) started to be generated on the specimen surfaces. The weight gains during the heating up period was hardly changed, indicating that the thickness of the titanium oxide scale was not influenced by heating temperature. Within the aluminisation duration, a triple-layered coating, composed of an outer layer of Al_3_Ti, a mid-layer of Al_2_Ti and an inner layer of AlTi developed. Interestingly, by prolonging the aluminising time, the outer layer of Al_3_Ti grew significantly whilst the growth of the mid-layer of Al_2_Ti and inner layer of AlTi was relatively suppressed. The growth of the aluminide coating followed a parabolic rate law after 1 h, which demonstrates that the aluminisation process was controlled by the diffusion of the elements.

Although the Ti and Ti-6Al-4V samples were heated at 150 °C for 4 h and flushed with inert gas, Ar, to remove as much air and moistures as possible, it is inevitable for certain amounts of air and moistures to remain in the furnace tube and crucible. Therefore, the highly reactive Ti specimens were rapidly oxidised during the heating period and a titanium oxide scale was generated. In general, an anatase scale forms at relatively low temperature (<718 °C) [25], and then a pure rutile scale develops, which is consistent with the current observation, that is, a rutile scale was identified between 800 and 900 °C by XRD analysis. With the generation of TiO_2_ scale on the sample surface, the residual oxygen in the crucible was gradually consumed. At the same time, the activator, NH_4_Cl, would discompose at about 520 °C during the heating up period, as follows:2NH_4_Cl = 2HCl + N_2_ + 3H_2_(1)

It was reported [9] that Ti could strongly react with oxygen when the temperature is over 550 °C. Therefore, the released gases from the dissociation of the activator at 520 °C might not be sufficient enough to develop aluminide and the formation of titanium oxide becomes inevitable. Nevertheless, the gases could surround the sample surface in the crucible and act as a barrier, thereby preventing the oxygen species from transporting to the sample surface and further reducing the oxygen partial pressure which is required to keep TiO_2_ remaining stable. Apparently, the lowest oxygen partial pressure would occur at the interface between the rutile scale and the substrate. The dissociation partial pressure of TiO_2_ increases when the experimental temperature arises. If there is no further oxygen species diffused into the interface zone, once the oxygen partial pressure is lower than the dissociation partial pressure of TiO_2_, the rutile becomes unstable and decomposes, thereby leading to the decline in oxygen content in the area close to the interface (Figure 6). The XRD reveals the low oxidation states of the compounds (e.g., Ti_1.86_O_3_, TiO_1.04_, TiO_0.48_), as shown in Figure 7. On the other hand, the rutile phase survives adjacent to the sample surface (Figure 6). When the temperature rises to 1000 °C, the rutile phase decomposes almost completely. As mentioned earlier, the thickness of the titanium oxide scale is barely increased during the heating up period (between 800–1000 °C), which further demonstrates the consumption of oxygen species around the sample due to the early stages of formation of the rutile scale.

During the heating up period, the following reaction occurs in the crucible:6HCl + 2Al = AlCl_3_ + 3H_2_(2)

The gaseous AlCl_3_ would transport to the sample surface. The complete dissociation of rutile at 1000 °C creates a condition that makes it possible for the reactions below to take place: AlCl_3_ = 3[Cl] + [Al](3)
[Al] + Ti = AlTi(4)

Indeed, the Ti mono aluminide (AlTi) was identified on the sample surface (Figure 7). The released Cl anions may react with Al in the crucible and make the aluminisation process self-catalytic in nature. When the aluminisation time increases, a full layer of AlTi forms, which results in the reduction of Ti activity on the sample surface. Meanwhile, the reaction (Equation (3)) would continue, leading to the rise of the Al activity on the sample surface. Apparently, Ti would diffuse outwards through the formed AlTi layer and interact with Al and make the following reaction happen:2[Al] + Ti = Al_2_Ti(5)

As a result, the mid Al_2_Ti layer develops. When the Al activity further builds up and the outward migration of Ti from the substrate slows down, the following reaction occurs:3[Al] + Ti = Al_3_Ti(6)

Consequently, the outer Al_3_Ti layer forms. At this point, a triple-layered coating is generated on the sample surface. It is clear that the coating grows primarily by the outward diffusion of Ti from the sample substrate, which explains the fact that the thicknesses of both the mid-layer of Al_2_Ti and inner layer of AlTi hardly increase within 10 h of aluminisation. After the triple-layered coating develops, the growth of the coating begins to be controlled by the outward transport of Ti from the sample substrate and obeys the parabolic rate law. It should be pointed out that the aluminide coating would not develop until the temperature rose to 1000 °C at which the TiO_2_ scale almost completely dissociates.

Slower growth of the aluminide coating on the Ti-6Al-4V alloy than on pure Ti was noted. This may attributed to two possible reasons: (i) the activity of Ti in the Ti-6Al-4V alloy is lower than that of Ti due to the addition of alloying elements, and (ii) the addition of Al and V probably slows the outward diffusion of Ti through the aluminide coating.

## 5. Conclusions

The commercially pure Ti and Ti-6Al-4V alloy were aluminised by the pack cementation process. The powder pack consisted of 4 wt% Al and 4 wt% NH_4_Cl balanced with Al_2_O_3_. The specimens were heated up to 1000 °C for up to 10 h. The following points are concluded:A TiO_2_ (rutile) scale around 20 μm thickness, rather than aluminide coating, developed during the period of heating up to 1000 °C. The thickness of the scale barely increased between 800 and 1000 °C.The rutile phase was reduced to the lower oxidation states when the experimental temperature arose. Almost all the rutile phase was dissociated at 1000 °C. Afterward, the aluminide coating began to develop.A triple-layered coating generated on both pure Ti and Ti-6Al-4V alloy and consisted of an outer layer of Al_3_Ti, a mid-layer of Al_2_Ti and an inner layer of AlTi. The thickness of the coating was dominated by the outer layer.The growth of triple-layered coating was controlled predominately by the outward diffusion of Ti from the sample substrates following the parabolic rate law.

## Figures and Tables

**Figure 1 materials-12-03097-f001:**
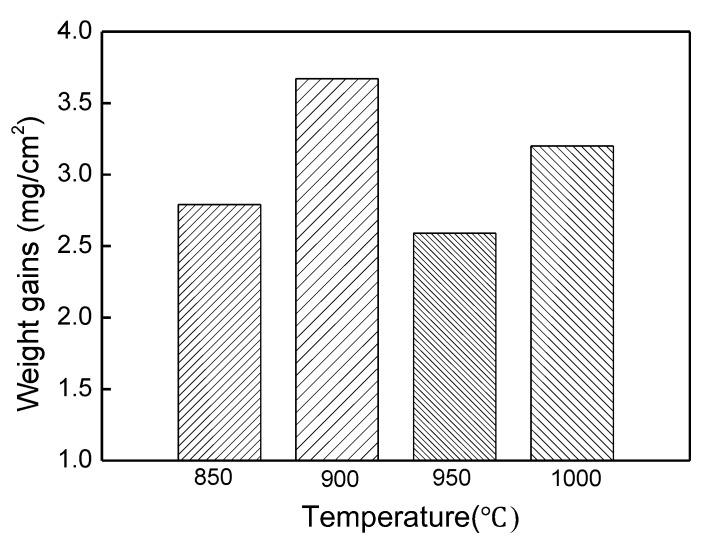
Weight gains of Ti heated up at 850–1000 °C in the aluminising pack for 0 h.

**Figure 2 materials-12-03097-f002:**
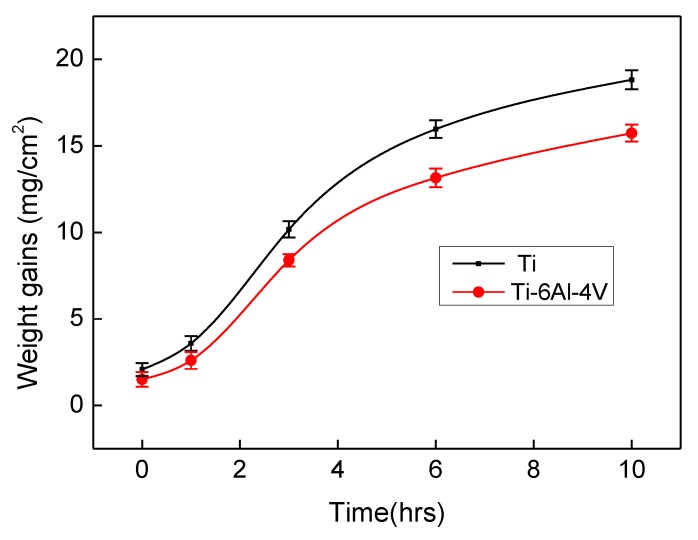
Weight gains of aluminised Ti and Ti-6Al-4V at 1000 °C for up to 10 h.

**Figure 3 materials-12-03097-f003:**
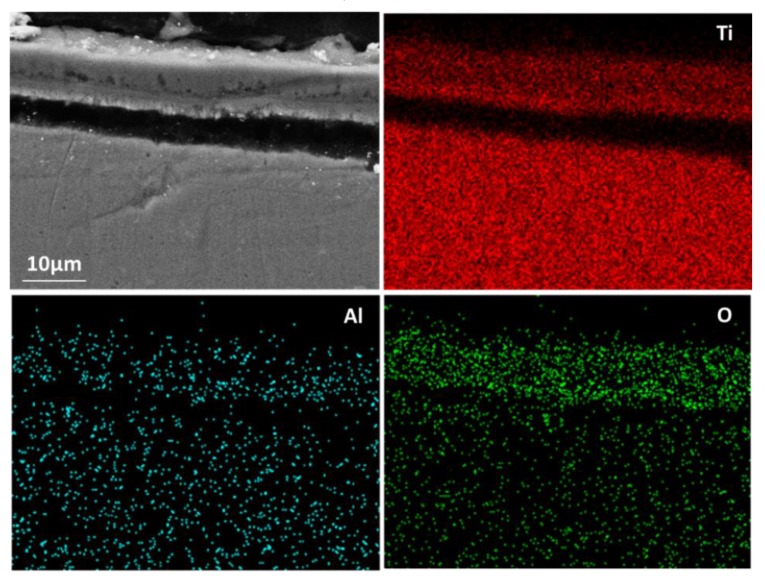
Scanning electron microscopy (SEM) image and energy dispersive spectrometer (EDS) mapping for cross-sectioned Ti heated up to 800 °C.

**Figure 4 materials-12-03097-f004:**
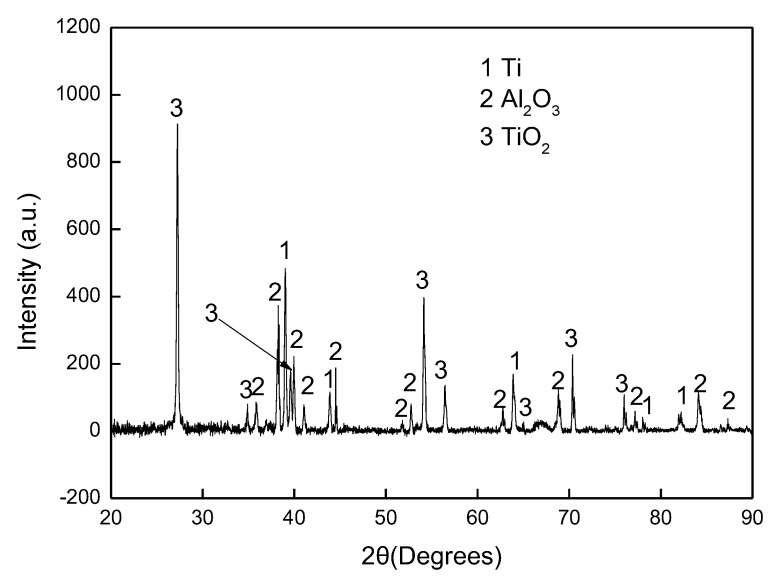
X-ray diffraction (XRD) pattern for cross-sectioned Ti heated up to 800 °C.

**Figure 5 materials-12-03097-f005:**
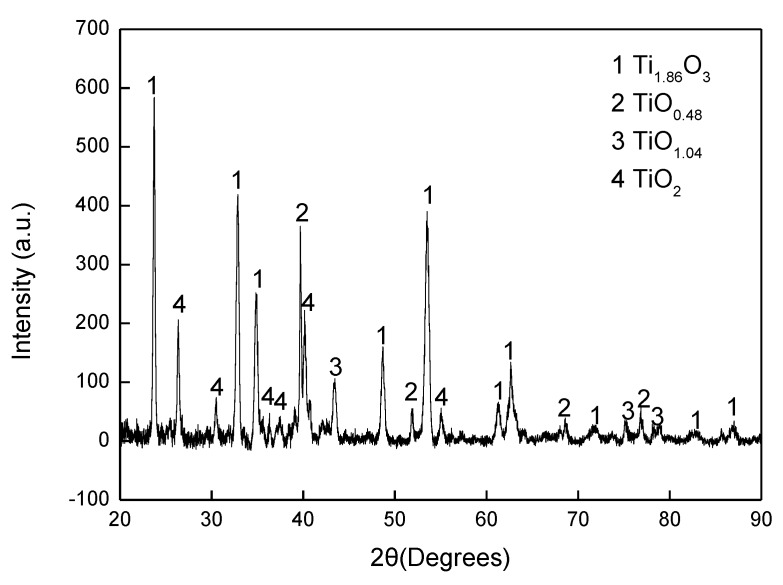
XRD pattern for cross-sectioned Ti heated up to 950 °C.

**Figure 6 materials-12-03097-f006:**
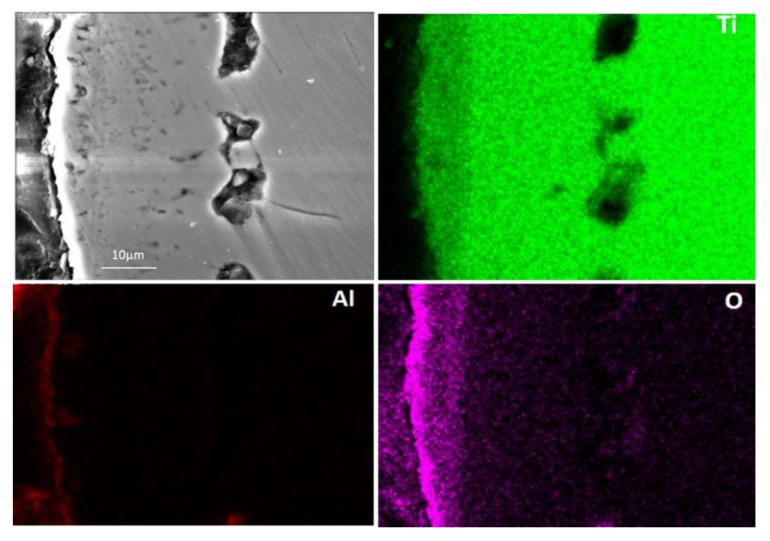
SEM image and EDS mapping for cross-sectioned Ti heated up to 950 °C.

**Figure 7 materials-12-03097-f007:**
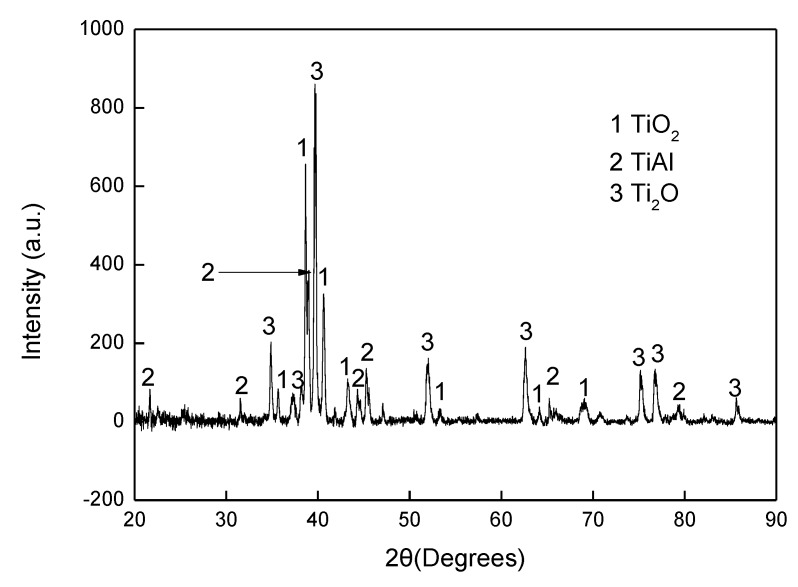
XRD pattern for cross-sectioned Ti heated up to 1000 °C.

**Figure 8 materials-12-03097-f008:**
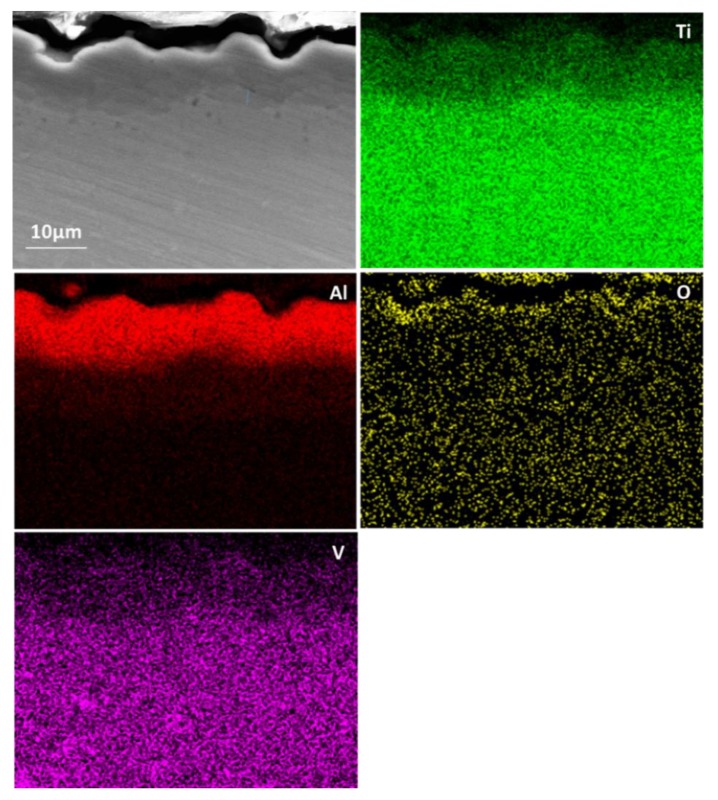
SEM image and EDS mapping for cross-sectioned Ti-6Al-4V aluminised at 1000 °C for 1 h.

**Figure 9 materials-12-03097-f009:**
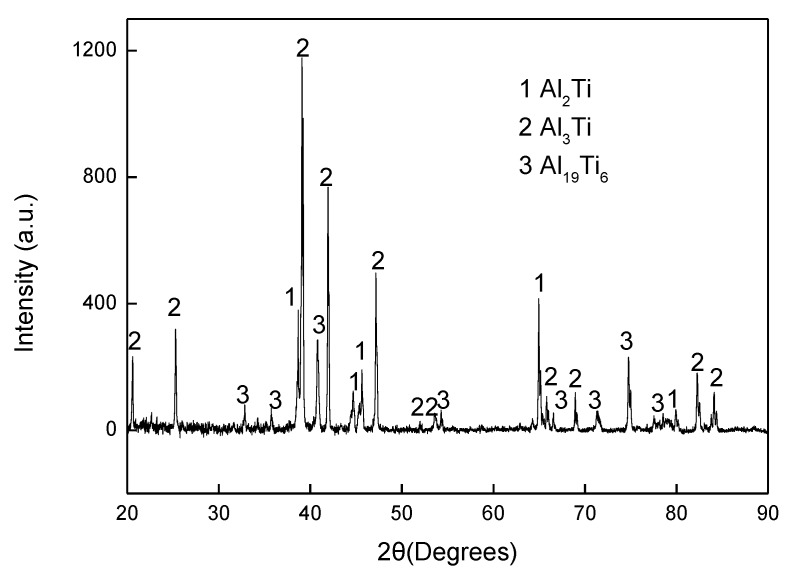
XRD pattern for cross-sectioned Ti-6Al-4V aluminised at 1000 °C for 1 h.

**Figure 10 materials-12-03097-f010:**
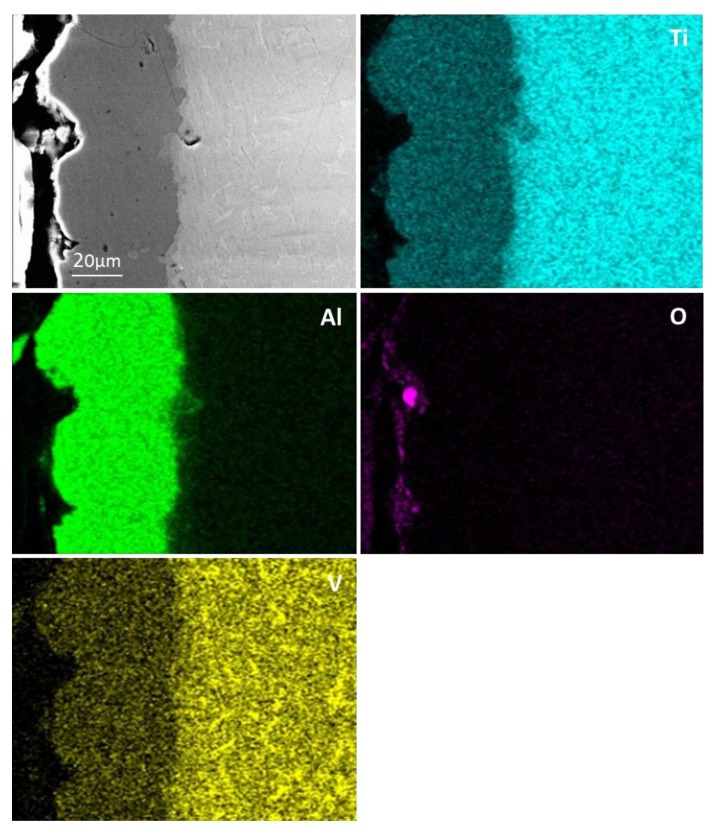
SEM image and EDS mapping for cross-sectioned Ti-6Al-4V aluminised at 1000 °C for 6 h.

**Figure 11 materials-12-03097-f011:**
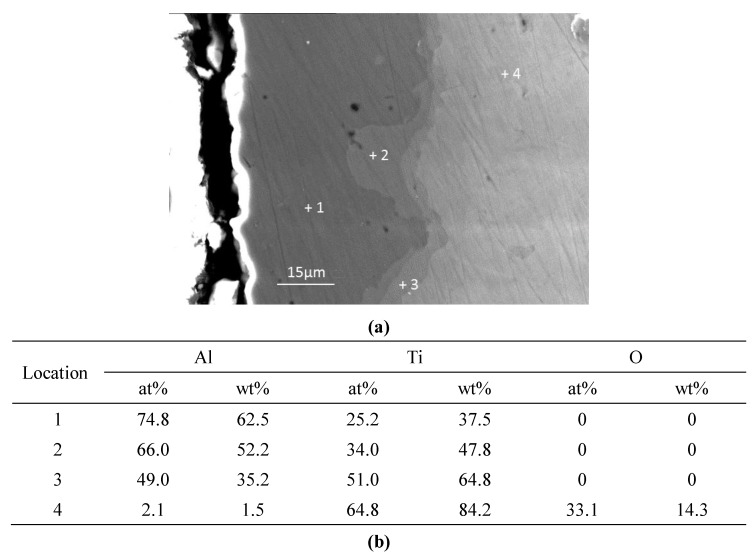
SEM image (**a**) and point quantitative elemental measurements (**b**) for cross-sectioned Ti aluminised at 1000 °C for 10 h.

**Figure 12 materials-12-03097-f012:**
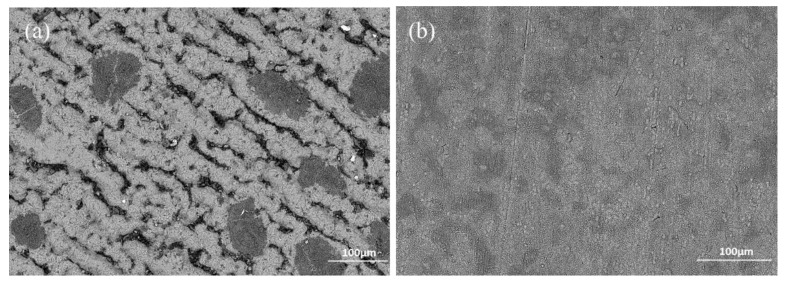
SEM surface morphologies for aluminised Ti and Ti-6Al-4V at 1000 °C for 6 h: (**a**) Ti and (**b**) Ti-6Al-4V.

**Table 1 materials-12-03097-t001:** Aluminising programme

Experiment	Temperature (°C)	Time (hours)
1	800	0
2	850	0
3	900	0
4	950	0
5	1000	0
6	1000	1
7	1000	3
8	1000	6
9	1000	10

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
