# Peer review of "Formation Mechanism of Aluminide Diffusion Coatings on Ti and Ti-6Al-4V Alloy at the Early Stages of Deposition by Pack Cementation"

_materials, 2019, doi:10.3390/ma12193097_

Round 1
Reviewer 1 Report
In this paper, the influence of temperatures at the early stages of coating development and the coating deposition kinetics were investigated in order to establish the formation mechanism of aluminide diffusion coatings on pure Ti and Ti-64 alloy by the pack cementation process, particularly at the early stages of aluminisation process. The research is interesting. However, to be accepted for publication the following comments need to be addressed.
1- The introduction section needs to be improved by referring to newer references and also citing related articles of the current journal.
2- The proficiency of the language needs a more improvement in the manuscript, for example not all, line 109 - weigh?
3- The dependences of the weight gains on both the temperature and time were obtained via weighting the samples. Thus, you are recommended to add the errors at each measuring point.
4- In characterisation section, how the samples were prepared for micrography? You are recommended to write the steps of preparing this sample.
5- The Font size be unified. Such as "Figure" and " Figure "
6- For measuring the thickness, did you measured them at different places then took the average? if yes, you are required to add the uncertainty of each value.
7- Figure 12 is needed to be improved, (scale bar).
8- The discussion section should be improved.
Author Response
Reviewer 1’s Comments:
In this paper, the influence of temperatures at the early stages of coating development and the coating deposition kinetics were investigated in order to establish the formation mechanism of aluminide diffusion coatings on pure Ti and Ti-64 alloy by the pack cementation process, particularly at the early stages of aluminisation process. The research is interesting. However, to be accepted for publication the following comments need to be addressed.
The introduction section needs to be improved by referring to newer references and also citing related articles of the current journal.
Reply – The Introduction section has been rewritten and more than 10 new references has been added and referred.
The proficiency of the language needs a more improvement in the manuscript, for example not all, line 109 - weigh?
Reply – English language has been significantly improved and the error that the reviewer pointed out has been corrected, as shown in the modified version.
The dependences of the weight gains on both the temperature and time were obtained via weighting the samples. Thus, you are recommended to add the errors at each measuring point.
Reply – The error bars have been added in Figure 2.
In characterisation section, how the samples were prepared for micrography? You are recommended to write the steps of preparing this sample.
Reply – The relevant information the Reviewer requested has been added in Experimental section.
The Font size be unified. Such as "Figure" and " Figure "
Reply – It has been corrected.
For measuring the thickness, did you measured them at different places then took the average? if yes, you are required to add the uncertainty of each value.
Reply – The uncertainties for the relevant values have been added.
Figure 12 is needed to be improved, (scale bar).
Reply – The scale bar has been redone.
The discussion section should be improved.
Reply - Modification of Discussion section has been carried out.
Reviewer 2 Report
There is a certain problem with the Introduction and references. In order to justify the statement “The experimental results reported in this paper appear to be tremendously different from the literatures, particularly, at the early stage of the aluminisation process” the Introduction should contain description of existing mechanism of pack oxidation of Ti and point out at missing (or not) description of early stage aluminisation peculiarities. Is it possible that at early stages of the process concentration of gaseous phase is not sufficient to form aluminide, TiO2 forms inevitably in the absence of reducing atmosphere and this is obvious? Novelty is not stated. There are a lot of works concerning such Ti treatment, such as Scripta Metallurgica Volume 23, Issue 5, May 1989, Pages 685-689 (reports Al3Ti/Al2Ti/AlTi structure); The influence of aluminizing process on the surface condition and oxidation resistance of Ti-45Al-8Nb-0.5(B, C) alloy Coatings 8(3),113; Oxidation of Metals Volume 58, Issue 1-2, August 2002, Pages 197-216 Effect of ternary elements on the oxidation behavior of aluminized TiAl alloys; Formation of Diffusion Aluminide Coatings on γ-TiAl Alloy with In-Pack and Out-Pack Processes; Pack deposition of coherent aluminide coatings on γ-TiAl for enhancing its high temperature oxidation resistance; Microstructure and high temperature oxidation resistance property of packing Al cementation on Ti-Al-Zr alloy; Effect of pre-oxidation on the properties of aluminide coating layers formed on Ti alloys and many, many others, but they are missing in the Ref list, Introduction, Discussion.Author Response
Reviewer 2’s Comments and Suggestions for Authors
There is a certain problem with the Introduction and references. In order to justify the statement “The experimental results reported in this paper appear to be tremendously different from the literatures, particularly, at the early stage of the aluminisation process” the Introduction should contain description of existing mechanism of pack oxidation of Ti and point out at missing (or not) description of early stage aluminisation peculiarities. Is it possible that at early stages of the process concentration of gaseous phase is not sufficient to form aluminide, TiO2 forms inevitably in the absence of reducing atmosphere and this is obvious? Novelty is not stated. There are a lot of works concerning such Ti treatment, such as Scripta Metallurgica Volume 23, Issue 5, May 1989, Pages 685-689 (reports Al3Ti/Al2Ti/AlTi structure); The influence of aluminizing process on the surface condition and oxidation resistance of Ti-45Al-8Nb-0.5(B, C) alloy Coatings 8(3),113; Oxidation of Metals Volume 58, Issue 1-2, August 2002, Pages 197-216 Effect of ternary elements on the oxidation behavior of aluminized TiAl alloys; Formation of Diffusion Aluminide Coatings on γ-TiAl Alloy with In-Pack and Out-Pack Processes; Pack deposition of coherent aluminide coatings on γ-TiAl for enhancing its high temperature oxidation resistance; Microstructure and high temperature oxidation resistance property of packing Al cementation on Ti-Al-Zr alloy; Effect of pre-oxidation on the properties of aluminide coating layers formed on Ti alloys and many, many others, but they are missing in the Ref list, Introduction, Discussion.
Reply – It is known from literatures that Ti could strongly react with oxygen when the temperature is over 550oC whilst the activator (NH4Cl) dissociates at about 520oC. The initial released gases from the dissociation of activator might not be sufficient enough to develop an aluminide coating to stop the formation of titanium oxide. This argument has been given in the Discussion section.
All literatures the reviewer gave were found, as shown below. Apart from this, more new literatures were searched. Correspondingly, the Introduction has been significantly improved.
Hiroshi Mabuchi; Tatsuya Asai; Yutaka Nakayama. Aluminide coatings on TiAi compound. Scripta Metallurgica. 1989, 23, 685-689.
Wojciech Szkliniarz, Moskal Grzegorz, Agnieszka Szkliniarz, Radosław Swadźba, The influence of aluminizing process on the surface condition and oxidation resistance of Ti-45Al-8Nb-0.5(B, C) alloy Coatings 8(3),113.
HwanGyo Jung; Kyoo Young Kim. Effect of ternary elements on the oxidation behaviour of aluminized TiAl alloys. Oxid. Met. 2002, 58, 197-216.
Nouri, S.; Rastegari, S.; Mirdamadi, Sh.; Hadavi, M. Formation of diffusion aluminide coatings on γ-TiAl alloy within-pack and out-pack processes. Trans. Indian Inst. Met. 2015, 68, 867–871.
Z.D. Xiang, S. Rose, P.K. Datta, Pack deposition of coherent aluminide coatings on γ-TiAl for enhancing its high temperature oxidation resistance; Surface and Coatings Technology, Volume 161, Issues 2–3, 2 December 2002, Pages 286-292,
Tian, K. Zhou, Y.C. Zou, H. Cai, Y.M. Wang, J.H. Ouyang, X.W. Li, Microstructure and high temperature oxidation resistance property of packing Al cementation on Ti-Al-Zr alloy; Surface and Coatings Technology, Volume 374, 25 September 2019, Pages 1051-1058.
G. Zhao, W. Zhou, Q.D. Qin, Y.H. Liang, Q.C. Jiang, Effect of pre-oxidation on the properties of aluminide coating layers formed on Ti alloys and many, Journal of Alloys and Compounds, Volume 391, Issues 1–2, 5 April 2005, Pages 136-140,
Round 2
Reviewer 1 Report
Thank you for considering my comments and recommendations in your revised version.
Reviewer 2 Report
In my opinion, the manuscript was improved and the aims were stated more correctly.